# A 53-µA-Quiescent 400-mA Load Demultiplexer Based CMOS Multi-Voltage Domain Low Dropout Regulator for RF Energy Harvester

**DOI:** 10.3390/mi14020379

**Published:** 2023-02-02

**Authors:** Balamahesn Poongan, Jagadheswaran Rajendran, Li Yizhi, Selvakumar Mariappan, Pharveen Parameswaran, Narendra Kumar, Masuri Othman, Arokia Nathan

**Affiliations:** 1Collaborative Microelectronic Design Excellence Center (CEDEC), Universiti Sains Malaysia, Bayan Lepas 11900, Penang, Malaysia; 2Department of Electrical Engineering, Faculty of Engineering, University of Malaya, Kuala Lumpur 50603, Wilayah Persekutuan, Malaysia; 3Institute of Microengineering and Nanoelectronics, Universiti Kebangsaan Malaysia, Bangi 43600, Selangor, Malaysia; 4Darwin College, Cambridge University, Cambridge CB3 9EU, UK

**Keywords:** multi-voltage domain low dropout, radio frequency energy harvester, power management unit, bandgap voltage reference, system on chip, internet of things, feedback network, demultiplexer

## Abstract

A low-power capacitorless demultiplexer-based multi-voltage domain low-dropout regulator (MVD-LDO) with 180 nm CMOS technology is proposed in this work. The MVD-LDO has a 1.5 V supply voltage headroom and regulates an output from four voltage domains ranging from 0.8 V to 1.4 V, with a high current efficiency of 99.98% with quiescent current of 53 µA with the aid of an integrated low-power demultiplexer controller which consumes only 68.85 pW. The fabricated chip has an area of 0.149 mm^2^ and can deliver up to 400 mA of current. The MVD-LDO’s line and load regulations are 1.85 mV/V and 0.0003 mV/mA for the low-output voltage domain and 3.53 mV/V and 0.079 mV/mA for the high-output voltage domain. The LDO consumes only 174.5 µW in standby mode, making it suitable for integrating with an RF energy harvester chip to power sensor nodes.

## 1. Introduction

A radio frequency energy harvester (RFEH)–based system on chip (SoC) is an integrated circuit that is capable of operating with much less dependency on battery energy [1]. It is made up of many different sub-circuit blocks, such as analog and mixed signal blocks, and each one requires a different voltage domain from the power source, which has high power quality requirements. Hence, RFEH can extend the life of battery-powered SoCs.

A wireless sensor node (WSN), on the other hand, which is made up of four major components—a sensing unit, a power unit, a processing unit, and a transmitter—requires a better power solution to extend the battery life of the sensor, which continuously monitors, captures, and transmits data to the processing unit for processing [2,3]. Each wireless sensor node component operates in a different voltage domain.

Besides this, a series of prominent microcontrollers widely used in the industry still require a dynamic voltage scaling power solution. The microcontrollers have a supply voltage range that varies based on the system frequency. For example, flash memory programming is not necessary for microcontrollers, such as MSP430G2001, when the system frequency is 1 MHz. During this phase, the supply voltage range is from 1.8 V to 3.6 V. Meanwhile, the supply voltage range is from 2.2 V to 3.6 V if flash memory programming is needed [2,4]. The static or single-power solution is not helpful in such conditions. A notable power solution is required for the system to increase its power efficiency.

A single-inductor multiple-output (SIMO) device that switches out multiple independent DC/DC converters could be the solution for multi-voltage domains [5,6,7]. The SIMO DC/DC converter can regulate multiple outputs while maintaining high power efficiency thanks to a single internal inductor component. The SIMO approach has an advantage over traditional DC/DC regulators in that it consumes less power and occupies less chip area than traditional methods, which necessitate the use of multiple independent regulators to supply multiple domains. Although this strategy helps to increase power efficiency, large off-chip inductors prevent its use in SoC architecture because they take up a lot of space. Aside from that, output ripples, switching noise, and cross regulation from the switching regulator, as well as the failure to provide a clean supply to the device, are significant flaws in this solution.

In the last decade, the power management integrated circuit (PMIC), consisting of a switching regulator and several low-dropout regulators, has sparked much interest as a multi-voltage domain solution. Even though switching regulators do not provide clean power, they are more efficient than LDOs in terms of the massive voltage drop from input to output. Because the switching regulator’s output must be ripple-free, the LDO is a good solution for providing clean power because of its high power supply rejection capability. Both regulators were built as a single integrated circuit to improve power efficiency and reduce ripple in the regulated voltage. LDOs are placed in the back end and switching regulators are placed in the front end. However, when used for an embedded or SoC solution, the complexity of fabricating inductors for PMIC is a significant disadvantage [8]. Furthermore, PMIC is better suited for high-power systems than low-power solutions.

Because LDO can provide a clean supply with a strong supply rejection ability, it is a popular alternative to PMIC for supplying different voltages for SoC system applications [9]. However, a large external capacitor was placed at the output LDO to reduce overshoot and undershoot while maintaining stability. This is difficult for SoC applications due to space constraints. The use of capacitorless, OCL-LDO to overcome area constraints is ideal for portable electronic devices and SoC power delivery applications. However, to support multiple voltage domain operations in SoC, typically multiple LDOs are needed to accommodate in the SoC, which consumes more area on the SoC chip. Furthermore, power efficiency in numerous independent LDO have deteriorated because all LDO consume power continuously, even if some voltage domains only operate infrequently and do not require constant supply [10].

In addition to multiple independent LDO, researchers are interested in designing LDO with programmable output voltage [11,12,13]. The output voltage was varied by changing the ratio of the feedback resistor via the transistor switch. The control signal activated and deactivated the transistor switch. The programmable output voltage topologies in the cited paper [11,12,13] used control logic, consisting of many CMOS switches in the feedback network, to vary the output voltage. Various topologies such as multi-voltage control, adaptive reference control, dynamic voltage, and frequency scaling were used to control the gate voltage of the CMOS switches to achieve the desired output voltage of LDO. These circuits consume a large area on the chip.

J. H. Wang et al. proposed a programmable 32-step output voltage that modifies the digital pulse width modulation (DPWM) clock and supplies it with a clean supply voltage [14]. The output voltage was modified using digital LDO topology in this technique. The phase frequency detector detected the clock differences between the reference clock and DPWM output clock. The output clock of DPWM was fed into the phase-frequency detector, PFD. When the output clock of DPWM differs from the reference clock in frequency or phase, the PFD captures it and generates a voltage signal to indicate the variation in phase or frequency. The main objective of the TDC employed in this design is to convert the output of the PFD signal into a 5-bit digital signal, which is then used to correct the output clock of the DPWM by adjusting the output voltage of the proposed LDO regulator, as explained in reference [14].

A higher resolution of step size is also necessary for DLDO to ensure that the output voltage is regulated with high accuracy. Moderate step size or step resolution leads to drawbacks in output voltage accuracy [14]. However, high power consumption and a large chip area are the trade-offs for the high-resolution step circuit. Hence, a technique for programmable reference voltage with a wider input common mode error amplifier has been introduced in addition to programmable output voltage LDO [15,16].

To address the issues above, the MVD-LDO has been proposed in this work. The proposed MVD-LDO has an on-chip capacitor and can handle up to 400 mA load current, making it suitable for embedded and SoC applications. Section 2 and Section 3 describe the application and circuit architecture of the proposed multi-voltage domain low dropout regulator with demultiplexer, followed by Section 4, which elaborates on circuit implementation of multi-voltage domain LDO. Section 5 displays the measurement results of the proposed design, and Section 6 presents the conclusion.

## 2. MVD-LDO for RFEH Based IoT Systems

The proposed MVD- LDO is suitable for RFEH-based IoT System on Chip (SoC). The SoC’s basic building blocks are processor units, sensors, transmitters, and receivers. Because the SoC contains numerous devices, different voltage rails were required to power the SoC. Even if each device serves a distinct purpose, it is not necessary to keep them all turned on at all times because this reduces battery life. As a result, MVD-LDO is a viable alternative to conventional multi-LDO, which consumes more power, costs more, and takes up more space. Figure 1 depicts a block diagram of an MVD-LDO for RFEH-based Internet of Things applications.

To supply the source, the MVD-LDO’s output is distributed to all the devices, and each device’s IO pin is connected to the processing unit to signal the device’s state while it is powered up. The processing unit will initially configure the LDO to output the voltage following the sequence. Once the work is completed, the first device sends an acknowledgment signal to the processor via an IO pin. In response to the acknowledgement from the previous device, the processor determines which device should be turned on next. As a result, the MVD-LDO is used to implement a power-efficient supply mechanism on the SoC.

## 3. Circuit Architecture of MVD-LDO

Figure 2 illustrates the schematic of the MVD-LDO which is capable of producing up to four regulated output voltages (Vout 1, 2, 3, 4) with a DEMUX controller.

This MVD- LDO regulator comprises a set of feedback resistors, a pass element, a low-power error amplifier, bandgap reference voltage (BGR), demultiplexer, and on-chip load capacitor. A resistive divider of the output voltage generates the feedback voltage, V_FB_, which varies proportionally with the output voltage. A reference voltage, V_REF_, is supplied through the bandgap reference (BGR) circuit. When the feedback voltage, V_FB_, deviates from the reference voltage, V_REF_, the error amplifier is used to correct the error at the output voltage by adjusting the gate voltage of the pass device. The error amplifier corrects errors by varying the pass transistor’s gate voltage and assuring the regulated voltage is within design parameters.

The feedback circuit in the proposed MVD-LDO consists of R_1_, R_2A_, R_2B_, R_2C_, and R_2D_ to provide four output voltages. In addition, for switching purposes, an NMOS transistor has been added to each pathway of the feedback resistor. In response to user input, the demultiplexer activated the gate voltage of an NMOS transistor. Only one output voltage can be regulated at any given time. The constituent units are discussed in the following sections.

## 4. Circuit Implementation of MVD-LDO

Figure 3 depicts a single voltage domain LDO with the output voltage controlled by feedback resistors or the reference voltage of the op-amp [17].

The LDO’s V_out_ is given as:(1)$$V_{out}=\left(1+\frac{R_{1}}{R_{2}}\right)V_{FB}$$
where V_FB_, is the feedback voltage of the single voltage domain LDO. The error amplifier regulates the gate voltage of the pass transistor by comparing V_FB_ to the reference voltage, V_REF_. If an error amplifier’s input is unequal, the pass transistor’s gate voltage is varied to control the required output voltage. This activity loops indefinitely to ensure that the output voltage is accurately regulated.

Off-chip placement of the feedback resistor is common, and the combination of feedback resistor values is positioned according to the desired regulated output voltage. Unfortunately, using an off-chip approach raises the design cost and is unsuitable for embedded design. Aside from that, an incorrect resistor value influences the regulator’s current efficiency, stability, and other critical parameters, resulting in a system with poor power efficiency.

Altering the reference voltage, V_REF_, while preserving the feedback resistor’s ratio is an additional way for altering the output voltage [17]. Since the reference voltage usually supplied from the bandgap reference, which is conspicuous across the process, voltage, and temperature, and tweaking the reference voltage is not a good solution to achieve acceptable output voltage. Hence, in MVD-LDO, adjusting the feedback resistor ratio has been adopted to govern the multi-voltage domain. Figure 4 displays the design of the MVD-feedback LDO’s resistor.

In our work, the bottom resistor of the feedback network circuit has been developed using a resistor bank, which increases the precision of the output voltage and offers multiple V_out_. Since this design intends to handle four V_out_, a total of four resistors, R_2A_, R_2B_, R_2C_, and R_2D_, have been added to the bottom of the feedback resistor. NMOS transistors MN_A_, MN_B_, MN_C_, and MN_D_ serve as switches. At a given moment, only one path is activated. In accordance with the ratio of the feedback resistor in the active circuit, the output voltage is regulated. An on-chip low power consumption demultiplexer design has been implemented to manage the switches and adjust the output voltage correspondingly. The power consumption of the DEMUX is only 68.85 pW. This is considered in the worst-case scenario, when all three inputs of the DEMUX have been turned on.

The digital 2:4 demultiplexer has been utilized on the proposed MVD-LDO. This demultiplexer uses three input signals, which are one enable pin and two logic input pins, to generate four output signals that regulate the feedback network’s switched-on state. A_0_, A_1_, A_2_, and A_3_ are the demultiplexer’s output signals, whereas V_A, V_B, and EN are the input signals

Similar to V_DD_, the input voltage of the demultiplexer is 1.5 V. The enable pin, EN, is employed to activate the LDO regulator. If the EN signal logic is LOW, the DEMUX is powered off as all four switches are turned off. Therefore, V_FB_ is the same as V_DD_. As a result, the gate voltage of the pass devices rises as the error amplifier’s output voltage increases, resulting in the eventual shutdown of the LDO. The shutdown process puts the SoC into sleep mode, critical for conserving battery energy.

To achieve low power consumption, the W/L of the demultiplexer’s transistors are optimized with reference to (2): (2)$$I_{D,sat}=\frac{{µ}_{p}.C_{ox}}{2}.\frac{W}{L}.{\left(V_{GS}-V_{T}\right)}^{2}$$

As a result, the demultiplexer consumes only 30.5 pA and 45.9 pA of current in standby and operating modes, respectively. The EN is used to toggle between operating and standby mode. The simulation results of the current consumption of the demultiplexer during standby and operating modes are shown in Figure 5. The simulation was performed on the Cadence Virtuoso platform with Silterra CMOS 180 nm process technology.

Most of the logic gate was shut down during standby mode since the EN signal is low. Therefore, it consumes minimal current. On the other hand, when all logic is enabled, the current consumption rises rapidly. In the proposed MVD-LDO, the EN, V_A, and V_B must be set to regulate the 1.4 V. This circumstance causes the DEMUX to consume more current.

Additionally, the MVD-LDO was also designed with high current efficiency. The current efficiency of the MVD-LDO is dependent on the load and quiescent current. Its current efficiency is expressed as (3): (3)$$CurrentEfficiency,\%=\frac{I_{\mathrm{load}}}{I_{load}+I_{bgr}+I_{EA}+I_{DEMUX}+I_{R}}\times 100\%$$
where I_load_ is the load current, I_bgr_ is the BGR current, I_EA_ is the error amplifier current, I_DEMUX_ is the demultiplexer current, and I_R_ is the current flowing into the feedback resistor bank. Optimizing the W/L of the transistors in the BGR, error amplifier, and the demultiplexer reduces the quiescent current without degrading other critical performance factors. I_R_ is reduced with reference to (4): (4)$$I_{R}=\frac{V_{REF}}{R}$$
where the V_REF_ is the reference voltage, and reduction of the quiescent current increases the current efficiency [18].

The simulated quiescent current of the MVD-LDO is plotted in Figure 6.

The MVD-LDO is designed to carry 400 mA of load current. To accomplish this, parallel PMOS transistors operating in the saturation region were implemented as pass transistors [19], as depicted in Figure 7. The size of the pass transistor is a matter to support the large current. The load current of the LDO is usually determined by the capacity of the pass device.

Smaller pass devices can only support a modest load current, whereas larger devices can support a large current based on the transistor’s capacity. By increasing the size of the transistor to support higher load current, it impacts the size of the chip. In the proposed MVD-LDO, multiple transistors are connected in parallel, while the size of each transistor is significantly smaller. The parallel pass transistor approach increases the capacity of the pass device to accommodate large load currents while keeping the transistor size smaller. The parallel PMOS also significantly reduces the quiescent current. Consequently, LDO’s current efficiency is enhanced.

The load regulation of the MVD-LDO is related to closed-loop DC output resistances, R_out,cl_, of the LDO, as illustrated in (5): (5)$$LR=\frac{\Delta V_{out}}{\Delta I_{out}}=R_{out,cl}\tilde{=}\frac{R_{out}}{1+\beta g_{mp}R_{out}A_{EA,O}}$$
where g_mp_ is the transconductance of the pass transistor, R_out_ is the output impedance, β is the feedback factor of the amplifier, and A_EA,O_ is the DC gain of the error amplifier.

To achieve good load regulation, the closed-loop DC output resistance, R_out,cl_ was designed to be smaller [20]. Referring to (9), the R_out,cl_, reduces when the DC gain of the error amplifier, A_EA,O_, increases. Figure 8 illustrates the high gain op-amp used as error amplifier in the MVD-LDO.

The high gain error amplifier consists of an operational transconductance amplifier as first stage and a common source amplifier as second stage. The common source delivers higher output voltage swing. M_1_ and M_2_ are the input stage which operates in the saturation region. The equivalent transfer function is given as:(6)$$H\left(s\right)=\frac{K(1+\frac{S}{\omega_{z}})}{\left(1+\frac{S}{\omega_{p1}}\right)\left(1+\frac{S}{\omega_{p2}}\right)}$$

The overall DC gain of the error amplifier is given as:(7)$$DCGain=gm_{M1}*gm_{M10}*\left(ro_{M2}//ro_{M9}\right)*\left(ro_{M5}//ro_{M10}\right)$$

The MVD-LDO achieves optimum load regulation and transient analysis by boosting the error amplifier’s DC gain. The transient analysis of the LDO is simulated and plotted in Figure 9.

Based on the transient analysis simulation, the output voltage ripple was obtained. It is evident that without the external capacitor, the output voltage variation is significantly lower. Figure 10, employed the simulation results of the output voltage ripple for all four outputs during full-load conditions.

## 5. Measurement Results

The proposed multi-voltage domain LDO regulator was fabricated using 180 nm CMOS process technology to verify the feasibility of the proposed design topology. Figure 11 depicts the micrograph of the proposed LDO.

The proposed LDO regulator can support a load current of up to 400 mA without using any external component, especially an off-chip capacitor (C_L_). Figure 12 illustrates the measured multi-voltage domain output voltage from 0.8 V to 1.4 V, with different configuration during no load conditions. The test configuration setup with digital logic is presented in Table 1.

The test configuration setup with digital logic is presented in Table 1.

Line regulation of LDO computes from the measured result from Figure 12 when the supply voltage is 1.5 V. For output voltage of 0.8 V, the line regulation is 1.85 mV/V, while for 1.0 V regulated voltage, the line regulation is 2.7 mV/V. Besides this, the line regulation values for output voltages of 1.2 V and 1.4 V are 2.19 mV/V and 3.53 mV/V, respectively.

The transient analysis of the measured waveform is shown in Figure 13. All four regulated output voltages look stable.

Figure 14 demonstrates the quiescent current of the multi-voltage domain LDO regulator for each test configuration. During this measurement, the regulator connects to zero load. Based on the measured results, the lowest quiescent current is 46.8 µA, while the highest is 52.9 µA with the output voltage of 0.8 V and 1.4 V, respectively. The highest multi-voltage domain output voltage contributes to the largest quiescent current, while otherwise, the relationship is reversed. The results dictate that output voltage is directly proportional to quiescent current. This quiescent current is the total operating current for bandgap reference, differential amplifier, and demultiplexer, which operate during no-load and full-load conditions.

The load regulation on the multi-voltage domains is shown in Figure 15. The output voltage was measured while varying the load from no-load condition to full-load condition with current step of 20 mA. The load regulation values for multi-voltages of 0.8 V, 1.0 V, 1.2 V, and 1.4 V are 0.0003 mV/mA, 0.000325 mV/mA, 0.0004 mV/mA, and 0.079 mV/mA, respectively. Like line regulation, the load regulation has the smallest load regulation when the lowest multi-voltage domain voltage is highest when the output voltage is huge.

Current efficiency is an important parameter in LDO regulators as the input and output current are close enough. To evaluate the current efficiency of the regulator, the input current and output current were measured during the full-load condition. Based on the measured results, the current efficiency of the regulator for all four output voltages is computed and displayed in Table 2.

The current efficiency of the MVD-LDO regulator was about 99.98% for all four tested configurations.

Table 3 summarizes the performance of the proposed multi-voltage domain LDO regulator and compares it with other state-of-the-art LDOs.

## 6. Conclusions

This paper proposes a capacitorless multi-voltage domain low-dropout regulator (MVD-LDO) in 180 nm CMOS. The MVD-LDO can operate at 1.5 V, encompass a broad output range of 0.8V to 1.2V, and provide 400 mA to each of its four output voltage domains. A low-power integrated demultiplexer realizes this efficient multiple output voltage. In addition, the MVD-LDO draws 46.83 µA for lower output and 52.88 µA for higher output in the absence of load. The design of the error amplifier was optimized with a high DC gain, resulting in outstanding load and line regulation of 0.001 mV/mA and 1.85 mV/V, respectively. With a load of 400 mA and across the entire voltage domain, the current efficiency was 99.98%, making this an efficient power management component unit for RF energy harvesters.

## Figures and Tables

**Figure 1 micromachines-14-00379-f001:**
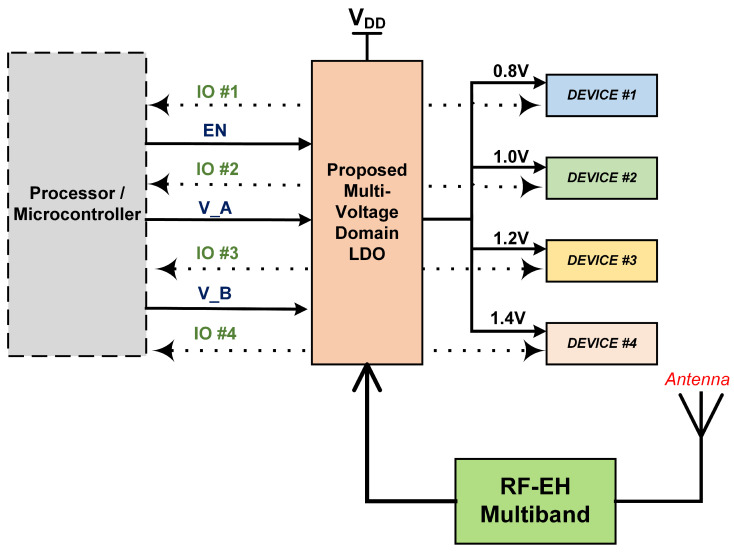
Block diagram of MVD-LDO for RFEH-based IoT applications.

**Figure 2 micromachines-14-00379-f002:**
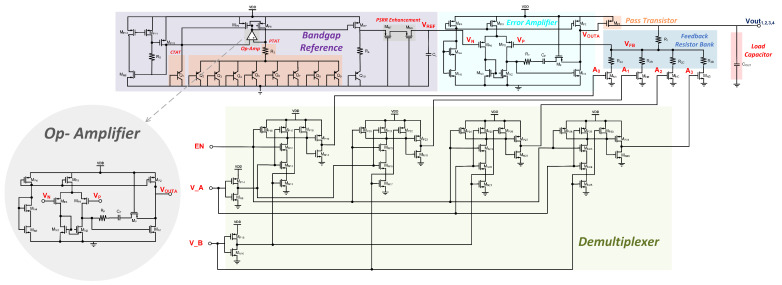
The circuit architecture of the MVD-LDO.

**Figure 3 micromachines-14-00379-f003:**
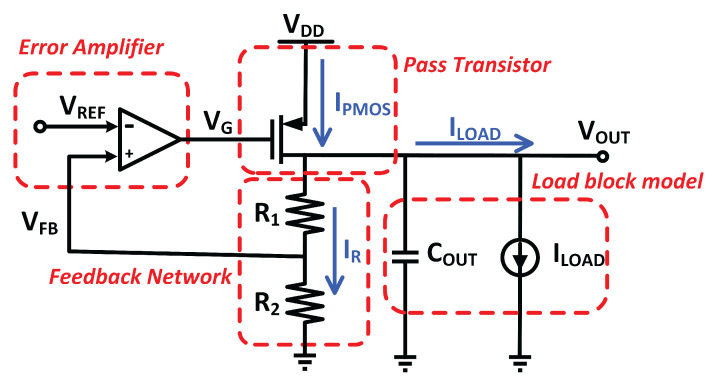
Conventional LDO regulator.

**Figure 4 micromachines-14-00379-f004:**
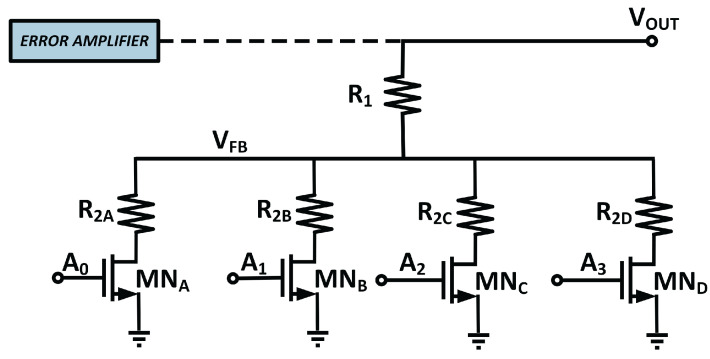
Feedback resistor design of MVD-LDO.

**Figure 5 micromachines-14-00379-f005:**
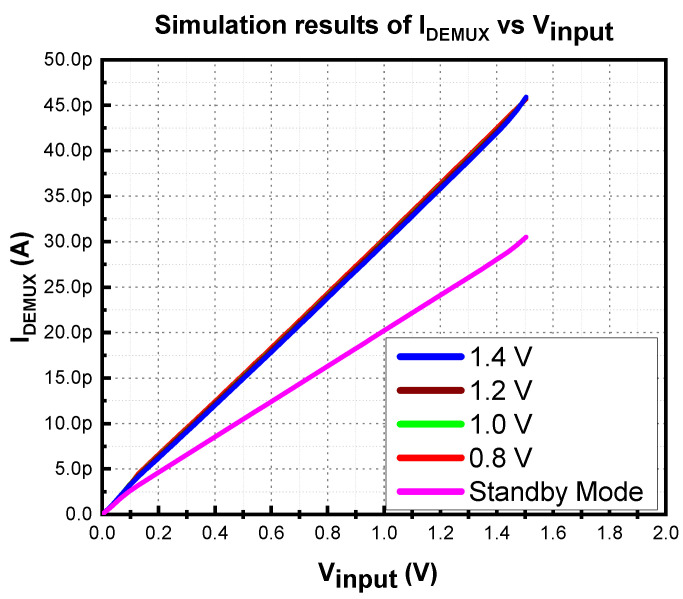
Simulation results of MVD-DEMUX’s current consumption.

**Figure 6 micromachines-14-00379-f006:**
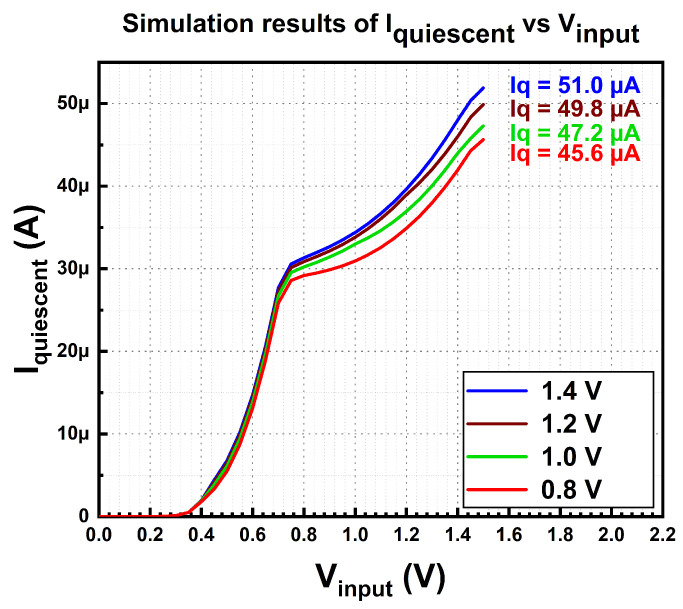
Simulation results of the MVD-LDO’s quiescent current.

**Figure 7 micromachines-14-00379-f007:**
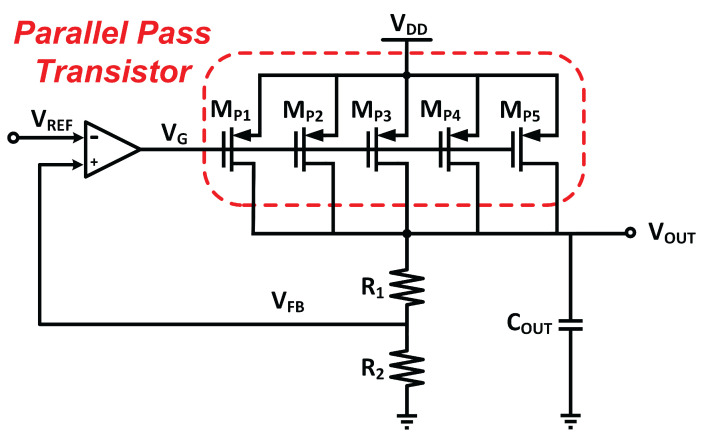
Parallel pass transistor design.

**Figure 8 micromachines-14-00379-f008:**
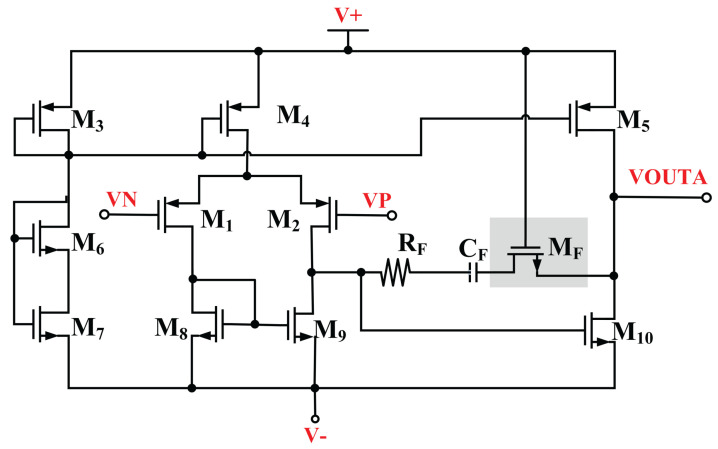
Schematic of the high gain error amplifier.

**Figure 9 micromachines-14-00379-f009:**
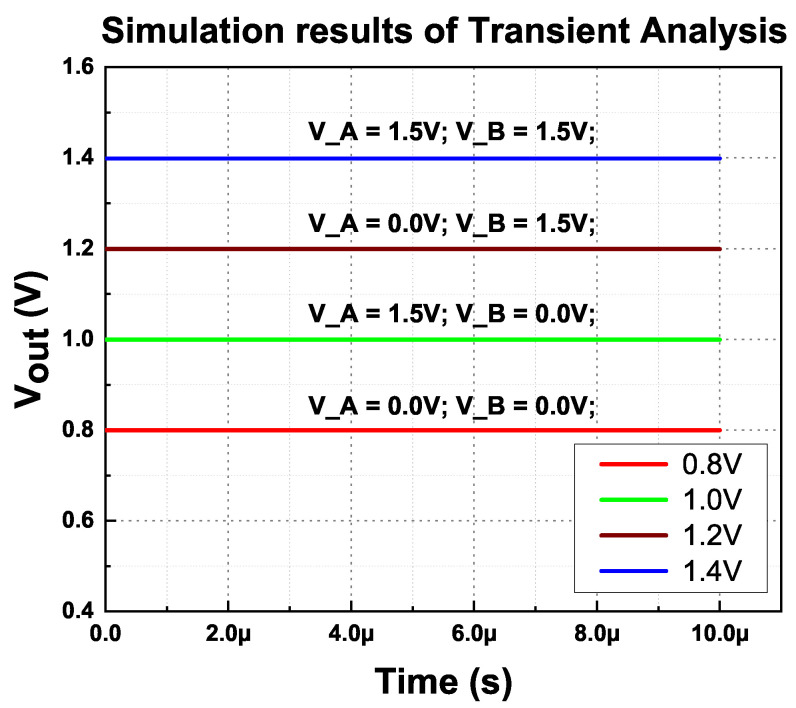
Simulation results of the MVD-LDO’s transient analysis.

**Figure 10 micromachines-14-00379-f010:**
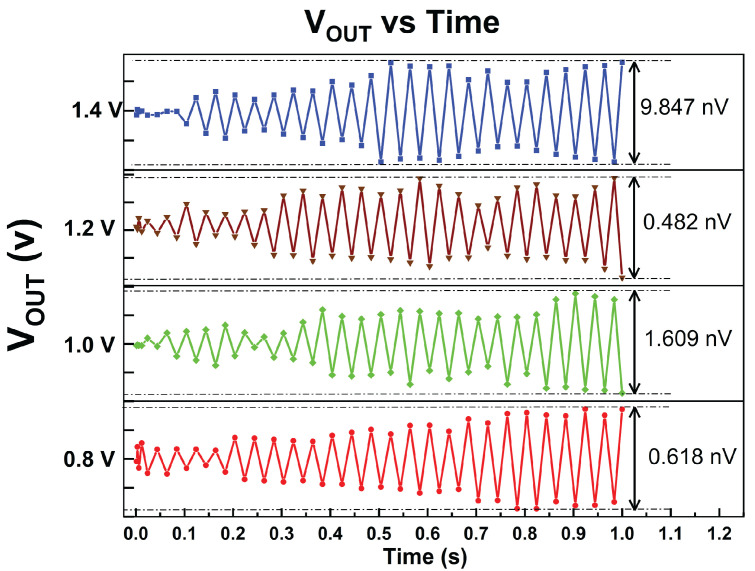
Simulated output voltage ripple of proposed MVD-LDO.

**Figure 11 micromachines-14-00379-f011:**
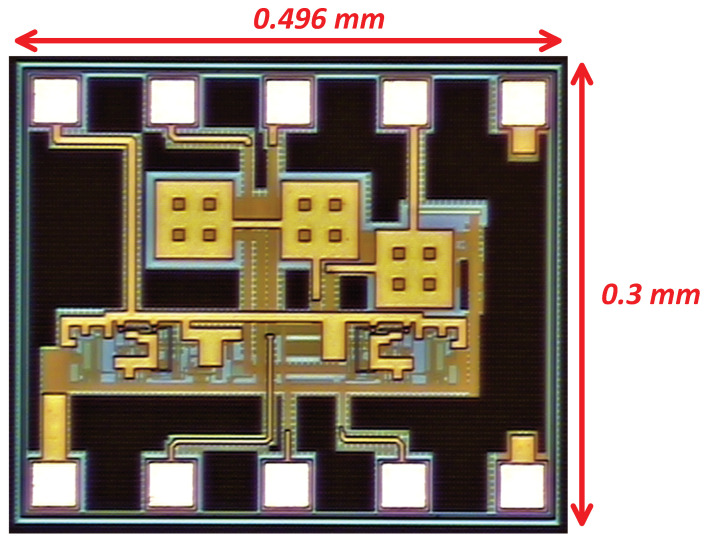
Micrograph of the proposed MVD-LDO.

**Figure 12 micromachines-14-00379-f012:**
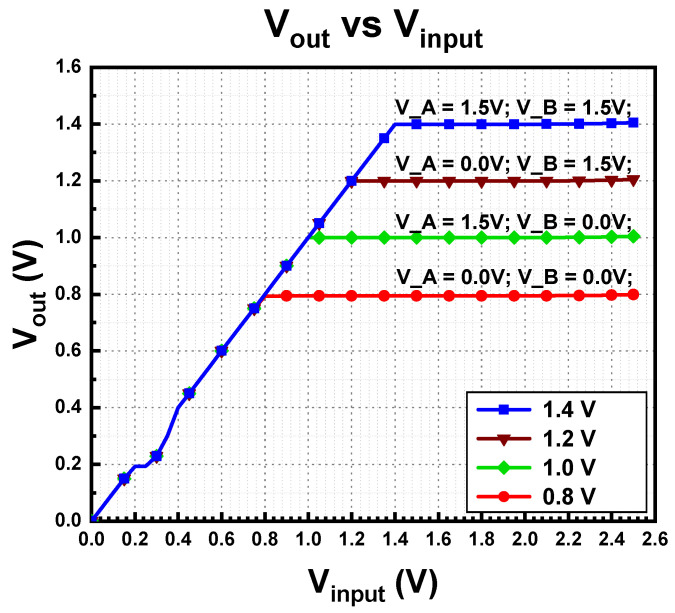
Output voltage of proposed MVD-LDO.

**Figure 13 micromachines-14-00379-f013:**
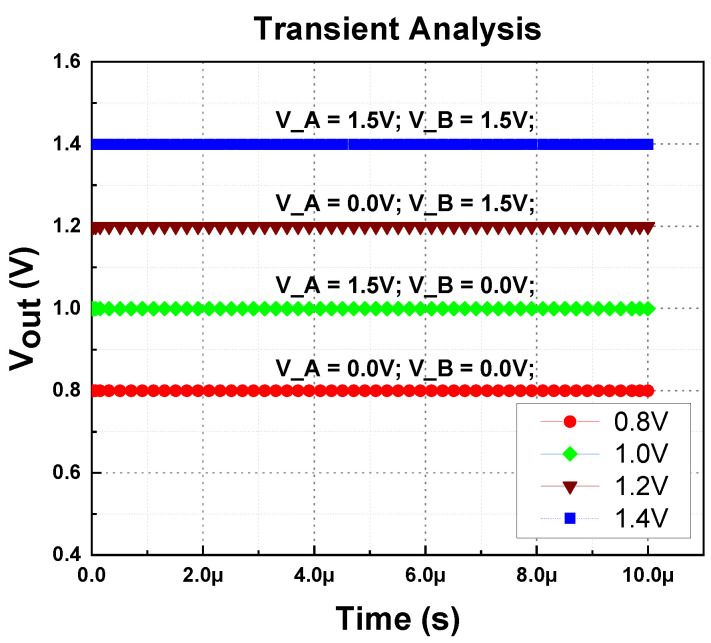
Transient analysis of proposed MVD-LDO.

**Figure 14 micromachines-14-00379-f014:**
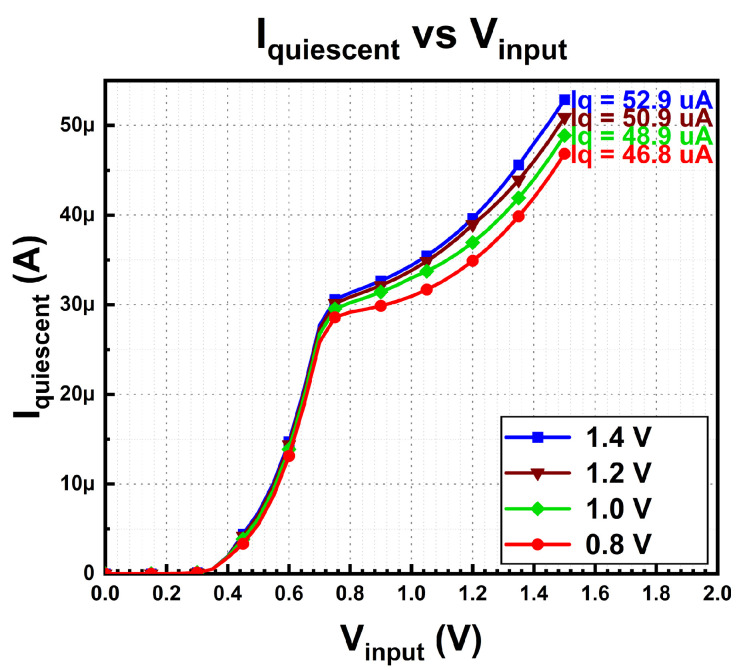
Quiescent current of proposed MVD-LDO.

**Figure 15 micromachines-14-00379-f015:**
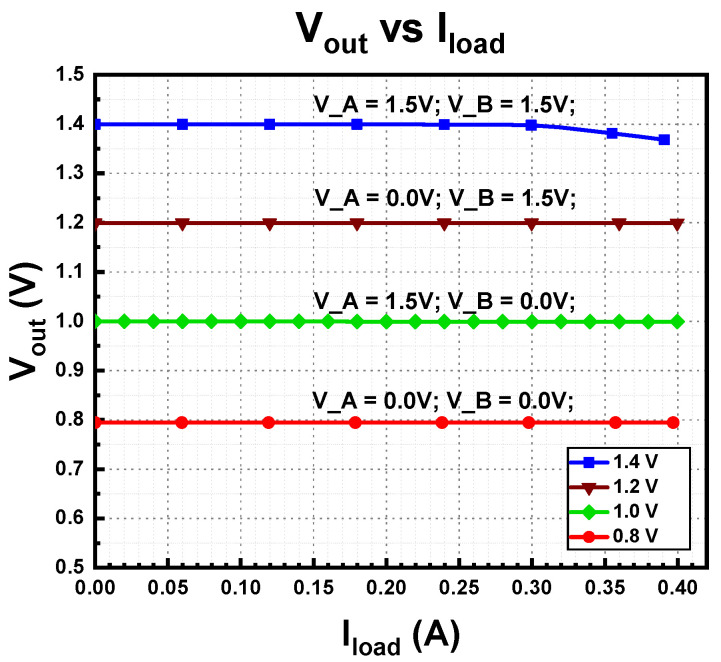
Measured load regulation of proposed MVD-LDO.

**Table 1 micromachines-14-00379-t001:** Test configuration of MVD-LDO regulator.

V_out_,V	V_A	V_B
0.8	Low	Low
1.0	Low	High
1.2	High	Low
1.4	High	High

**Table 2 micromachines-14-00379-t002:** Current efficiency of proposed MVD-LDO.

V_output_,V	I_output_,mA	I_input_,mA	Power Efficiency, %
0.8	399.5	399.6	99.98
1.0	399.5	399.8	99.98
1.2	399.7	399.8	99.98
1.4	397.0	397.1	99.98

**Table 3 micromachines-14-00379-t003:** Summarized performance of proposed MVD-LDO compared with other state-of-the-art LDOs.

Reference	[8]	[21]	[22]	[23]	[24]	This Work
Year	2021	2017	2020	2020	2019	2022
Topology	Multi Domain LDO	LDO	LDO	LDO	DLDO	Multi Domain LDO
Technology (nm)	180	130	40	180	65	180
Chip Area (mm^2^)	0.49	0.1825	None	0.037	None	0.149
Input Voltage, V_IN_(V)	3.3–3.6	1.05–2.0	1.1–1.9	1.4–1.8	0.8–1.2	1.5
Output Voltage, V_IN_(V)	3.2 (2.4, 1.6,0.8) ^1^	1.0	0.2–1.1	1.2–1.6	0.6–1.15	(0.8;1.0;1.2;1.4) ^1^
Dropout voltage, V_DO_(mV)	100	29.7	200	200	200	100–700
Load Current, I_load_(mA)	50	300	100	300	50	400
Quiescent Current, I_q_(µA)	239	14–120	56	0.94–255	26.25–105	46.83–52.88
Current efficiency, %	96.5	99.96	99.94	99.92	None	99.98
Load regulation (mV/mA)	(V_OUTH_: 4.38 × 10^−7^; V_OUTL_: 3.13 × 10^−6^) ^2^;	0.006	0.176 @ V_OUT_ = 0.2; 0.2 @ V_OUT_ = 1.1	0.1	None	0.0003–0.079;
Line regulation (mV/V)	1.13	0.44	0.857@ V_OUT_ = 0.2; 5.0 @ V_OUT_ = 1.1	5.33	None	1.85–3.53
Output Capacitor, C_O_ (µF)	None	1	1	1	0.1	None

^1^ Multi-voltage domain. ^2^ Unit for load regulation is %/mA.

## Data Availability

Not applicable.

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
