# Peer review of "A 53-µA-Quiescent 400-mA Load Demultiplexer Based CMOS Multi-Voltage Domain Low Dropout Regulator for RF Energy Harvester"

_micromachines, 2023, doi:10.3390/mi14020379_

Round 1

Reviewer 1 Report

Dear Authors,

please find hereafter a list of suggestion to improve your paper:

"...and the quantity of  LDO increases as space consumption gradually increases."

-> please explain what do you mean

"... The complicated programmable controller consumes
more power and takes up more chip area, which serves as disadvantage of this technique."

-> it is not clear what is the controller you are talking about

"...and sent the variation to the time-to-digital converter, decoded it into a
5-bit digital code."

-> This part is really hard to follow. Please explain more in details

"Furthermore, the complicated switching array has few tuning steps, which reduces 74
the voltage scaling’s effectiveness. As a result, both the area and the energy consumption 75
grow."

-> same comment as before

The paper is too long. Fig.5,6,7 can be omitted, together with the pertaining text, as well as Table 1-

quality of fig.14 is out of focus and it should be improved

The transient response of the measured waveform employed in Figure 16. The all four 235
regulated output voltage looks stable. 236

-> I do not see any transient... Please explain in more detail

Author Response

Dear Professor,

Thank you very much for all the constructive comments which improves the quality of the paper. We have addressed all of them and corrected the manuscript accordingly.

Thank you very much.

Dr Jagadheswaran Rajendran

on behalf of all the authors

Reviewer 2 Report

The authors need to describe the simulation process in detail so that the reader can repeat.

Author Response

(The authors gave the same response as above.)

Reviewer 3 Report

In this manuscript, a low power capacitorless demultiplexer based MVD-LDO is proposed. Its application and circuit architecture are presented.  The measurement results of the design are acceptable and efficient for RF energy harvesters. The manuscript can be accepted for publication after the following points were further addressed.  

1) Many syntax errors.

2) The references [17-19] has not been cited in the manuscript.

3) Figure 14 is not that clear. Hopefully the image sharpness will be higher.

Author Response

(The authors gave the same response as above.)

Reviewer 4 Report

This article is relatively well written. The method is relatively simple, but the target problem is practical. The methods and results of the experiment are clear. Logical flow are relatively smooth.

In the background introduction section, the author can go further into the practical application of this problem in industry. 

Author Response

(The authors gave the same response as above.)

Round 2

Reviewer 1 Report

Dear Authors,

I am quite not not fully satisfied  by this new revision.

Fig.5, 6 and 7 are useless, they can be found in any digital electronics textbook. The same applies to Table 1. There is no need to have them all in this paper.

The sentences:

"Based on cited paper [10], the control logic of LDO was controlled by an external controller such as SoC. While in [11] and [12], new topology was introduced to control the output voltage, such as multi-voltage control, adaptive reference control, and Dynamic voltage and frequency scaling. "

are quite weird. I do not understand what is the message.

"The clock difference is sent to the time-to-digital converter, TDC, to convert into a 5-bit digital code that will be used to adjust the output voltage to correct the output clock of DPWM."

What do you mean with "clock difference" ? A TDC needs a start and a stop. Who are they?

"In DLDO, a complex switching array requires high resolution tuning step to a regulated desired output voltage. Tuning steps which are meant in this sentence, is step resolution. Lower step resolution will reduce the voltage scaling effectiveness and generate low accuracy. A complicated design is required to create high step resolution, leading to higher area and power consumption of the LDO."

I am sorry to say I do not get the point....

fig.14 is out of focus and it should be improved

Fig.14 quality has not be improved. As it is, it could show almost anything. As it is, it could not appear on a scientific paper

Author Response

Dear Professor,

Thank you very much for the constructive comments. We have corrected the manuscript as per your advice.

Thank you very much

Best Regards,

Jagadheswaran Rajendran

on behalf of all authors

Round 3

Reviewer 1 Report

Dear Authors,

thank you for the update and the new version